# Exploring the Correlation between Time Management, the Mediterranean Diet, and Physical Activity: A Comparative Study between Spanish and Romanian University Students

**DOI:** 10.3390/ijerph19052554

**Published:** 2022-02-23

**Authors:** Elena-Simona Indreica, Georgian Badicu, Hadi Nobari

**Affiliations:** 1Department of Psychology, Education and Teacher Training, Faculty of Psychology and Education Sciences, Transilvania University of Brasov, 500068 Brasov, Romania; elena.indreica@unitbv.ro; 2Department of Physical Education and Special Motricity, Faculty of Physical Education and Mountain Sports, Transilvania University of Brasov, 500068 Brasov, Romania; 3Department of Physiology, School of Sport Sciences, University of Extremadura, 10003 Cáceres, Spain; hadi.nobari1@gmail.com; 4Department of Exercise Physiology, Faculty of Educational Sciences and Psychology, University of Mohaghegh Ardabili, Ardabil 56199-11367, Iran

**Keywords:** nutrition, physical fitness, food habits, time management, quality of life

## Abstract

Background: The investigation started from the premise that there are links between time management (TM), physical activity (PA), and the Mediterranean diet (MD). The aim of our study is to determine the correlation between the variables TM, the MD, and PA in Spanish and Romanian university students. Methods: The study was conducted on a group of 296 students (198 Romanian and 98 Spanish students between the ages of 23.44 ± 5.92 years, with 108 males and 188 females, where 171 were from the sports faculty and 125 were from the psychology faculty), using the Time Management Questionnaire (TMQ) to measure TM, the KIDMED test to measure MD, and the International Physical Activity Questionnaire-Short Form (IPAQ-SF) to measure PA. Results: The obtained results show that there are significant differences, regarding MD and PA, between the Romanian and Spanish respondents, between the respondents from the sports faculty and those from the psychology faculty, and between the female and male respondents. The TM variable did not show a significant difference depending on the country, faculty, or gender. There was only a significant relationship between the MD and the other two variables (TM and PA). Conclusions: Significant differences were observed between the variables the MD and PA, but not TM, depending on the country, gender, and faculty. There was only a significant correlation between the MD and the other two variables (TM and PA). The results provided us with relevant data for the need for a TM protocol to improve the MD adherence and PA in university students.

## 1. Introduction

Following the investigations carried out in recent years [1,2,3], we found that there is a link between time management and a number of variables aimed at the quality of life. We extended the area of interest to include the connection between these variables, physical activity (PA), and the Mediterranean diet (MD). Time management focuses on both the ability to effectively organize work tasks and the ability to adapt the time resources to optimize the quality of life [1,2,3]. In a previous study [1], we experimented with an individualized time management program related to learning styles. Currently, due to the need for online education, time management is affected, in this segment, by physical activities. Few studies link healthy food [2] to time management, and research between leisure time and physical activity is not found. A comparative study between Romanian and Spanish students would provide us with the relevant data on the need to use time management strategies and its relationship with PA, as well as relevant data on the MD.

As we know, PA and a healthy diet are key factors for avoiding major noncommunicable diseases. Moreover, according to the World Health Organization (WHO), the recommendation for adults to improve health parameters is to perform at least 150 min of moderate-intensity, or 75 min of vigorous-intensity, aerobic PA throughout the week (or an equivalent combination of moderate- and vigorous-intensity activities) [4]. On the other hand, diet habits have increasingly become a cause of concern because of their potential consequences for health [5,6]. In this way, the MD has been linked to a number of health benefits, including reduced mortality and the incidence of cardiovascular disease [7], as well as a relatively high level of self-rated health [7]. The particularities of the MD not only include a wide variety of foods, but also include the amount and consumption of these foods that is recommended. However, a particular emphasis is placed on the importance of food preparation [8,9].

This diet is abundant in minimally-processed plant-based foods, is rich in monounsaturated fat from olive oil, and is lower in saturated fat, meats, and dairy products. Further, a healthy diet is also positively related to a potentially reduced risk of infection and symptom severity in relation to COVID-19 [10,11].

Finally, when comparing Spain and Romania regarding the MD, we can say that, in Romania, this diet is little-used [12], the same being true for students. Additionally, in Spain, the MD is not of scientific journals or weight loss websites; it is simply a part of normal life.

Today, there are many studies on the relationship between PA [13] and the MD [14,15,16,17], but studies between leisure time and the two variables are not found.

Related to the association between the two variables, i.e., MD and PA, a study by Iaccarino et al. [18] showed that there is an association between PA and adherence to the MD, while they also have an inverse correlation with sedentary behavior [18]. Moreover, a systematic review with a meta-analysis by Malakou et al. [19] found that the promotion of the combination of the MD and PA showed a significant metabolic risk reduction [19].

Particularly in university, people tend to have a more sedentary life, due to their studies. Eating habits are another health factor to consider [20]. Recent studies show that students do not meet the general recommendations in relation to PA levels [21]. Following these studies, it has been concluded that students were less active than children and teenagers. Pavicic et al. [22] concluded that a high level of PA was associated with the MD, but not in a significant manner [22]. The variables associated with a high level of PA were a male gender, a younger age, a normal WHR (waist-to-hip ratio), a non-sedentary occupation, and a reduced sitting time [22].

In accordance with those presented, the investigation started from the premise that there are links between time management (TM), physical activity (PA), and the Mediterranean diet (MD).

Moreover, the present study aimed to conduct a comparative study between the sports faculty students and the psychology and education sciences faculty students, from Romania and Spain. We hypothesize the following for our study:

**Hypothesis** **1** **(H1).**
*There are differences between the variables TM, MD, and PA, depending on the country, faculty, and gender.*


**Hypothesis** **2** **(H2).**
*There is a relationship between the variables TM, MD, and PA.*


**Hypothesis** **3** **(H3).**
*The results for TM, MD, and PA are determined by the country of origin.*


## 2. Materials and Methods

### 2.1. Participants and Design

This non-experimental, cross-sectional, and descriptive study was conducted on a sample of 296 university students from Romania (66.9%; *n* = 198) and Spain (33.1%; *n* = 98). The sample was selected based on a unique criterion, including being a student at the Faculty of Psychology and Educational Sciences, or a student at the Faculty of Physical Education and Sports during the academic year 2020–2021. Respondents from the Faculty of Physical Education (57.8%; *n* = 171) and Psychology and Educational Sciences (42.2%; *n* = 125) were aged between 18 and 52 years old (23.44 ± 5.92 years old). The respondents participated voluntarily after receiving a detailed explanation of the objectives and nature of the study. The distribution of the questionnaires was done exclusively online. The questionnaires were written in Google Forms and were distributed on social networks, where 303 people answered. In this study, we excluded from the analysis 7 students that did not complete the inclusion criteria correctly (i.e., incomplete questionnaires or did not hand informed consent forms). There were a few deviations, as the items in the questionnaires were mostly set in the electronic system as mandatory. Thus, the final sample was comprised of 296 subjects, including108 males (36.5%) and 188 females (63.5%). There were 206 (69.6%) respondents from urban areas and 90 (30.4%) respondents from rural areas, and there were 171 (57.77%) students from the physical education and sports faculty and 125 (42.23%) students from psychology and educational sciences faculty. The students from Romania were enrolled in the Transylvania University of Brasov (*n* = 198), while the Spanish students were enrolled in the University of Extremadura (*n* = 98).

### 2.2. Time Management Questionnaire

The Time Management Questionnaire (TMQ) [1,2,3] contains 24 items. The 24 items are focused on three variables of the time management dimension: activity planning (AP), activity monitoring (AM), and streamlining activities (SA). Each dimension was measured by means of indicators and each indicator was represented in the questionnaire by one or more items. For the AP, the indicators are: making a schedule (items 5 and 19), prioritizing activities (items 1, 2, 4, and 9), time distribution (items 6 and 12), and clearly defined goals (items 10 and 11). For the AM, the indicators are: using a schedule (item 20), observing the time allotted to each activity (items 7, 8, 15, and 23), and redistributing time (items 13 and 16). For the SA indicators: achieving the activities (items 3, 21, and 22), asking for support (items 14 and 24), and making use of free slots of time (items 17 and 18). The answers are presented on a Likert scale from 1 (very rarely) to 5 (very often). For AP, AM, and SA, the indicator scores are summed. Based on the total scores of the three dimensions, the time management level is established: a score of 0–30 indicates non-existent TM; 31–60 indicates a TM deficit; 61–80 indicates moderate TM; 80–100 indicates good TM; and 101–120 indicates excellent TM. For the group in the present study, the Cronbach’s alpha coefficient is 0.84, which means that the test has a good internal consistency, as mentioned by other studies [1,2,3].

### 2.3. Mediterranean Diet (KIDMED Test)

The KIDMED test is a tool to evaluate the adherence to the MD for youths and children. The final score can include values between −4 and 12 points [23] and is based on a 16-question test that can be self-administered or conducted by an interview (with a dietitian, pediatrician, etc.). The questions denoting a negative connotation, with respect to the MD, are assigned a value of −1 (items 6, 12, 14, and 16 for the answer ‘yes’), and those with a positive aspect are assigned a value of +1 (the rest of the items). ‘No’ answers are given a 0. Moreover, the sums of the values from the administered test are classified into three levels: (1) >8, optimal MD; (2) 4–7, improvement needed to adjust the intake to Mediterranean patterns; and (3) ≤3, very low diet quality [24,25]. This test was developed and validated by Serra-Majem et al. [25].

### 2.4. Physical Activity Measures

For measuring the PA level, we used the official version of the International PA Questionnaire-Short Form (IPAQ-SF) [26,27]. This evaluation tool has, in its content, seven generic items. Our measurement evaluates the varying levels of the PA intensity, and the daily sitting time. This research considers that PA intensity, along with daily sitting time, as well as estimates the total amount of PA in MET-min/week and the time spent sitting. Moreover, the IPAQ-SF states that there are three categories of PA levels: “low”, “moderate”, and “high” [28,29]. The total weekly PA, expressed as MET-minutes per week (MET-min/wk^−1^), was calculated as follows: duration × frequency per week × MET intensity. Additionally, it was represented and summarized by walking, moderate-intensity PA, and vigorous-intensity PA for the week [29]. It is very important to mention the fact that all the questions are related to activities performed during the previous seven days [30]. The properties of the IPAQ-SF are appropriate for assessing the levels of PA in 18- to 65-year-old adults, in various environments [31,32]. From this study, the reliability of the IPAQ-SF was satisfactory (Cronbach’s α = 0.80) [32].

### 2.5. Procedure

First, we requested the approval of the study by the Ethics Committee of the University of Extremadura, which was granted with the code “641/CEIH/2018”. In addition, the informed consent of the respondents was requested through a document in which the nature of the study was detailed. In this study, we used the Google Forms to draw the questionnaires and we shared and collected the results from students by email. Participation in the study was voluntary, the online environment giving them the comfort of answering whenever they wanted, without having a time limit. The average time to complete the questionnaires was 15 min. Data collection took place over three months (July to September 2021), with access to questionnaires being closed at the end of September. The questionnaires were applied in the respondents’ mother tongue, Romanian or Spanish, in order to ensure the most correct decoding of the contents of the items. The questionnaires were also used in these languages in other studies [2,6,7,8,9,15,26,27,28,29]. The questionnaires were accompanied by clarifications regarding the purpose of the investigation and the protection of personal data.

### 2.6. Data Analysis

The data analysis was performed using IBM SPSS^®^ Statistics Version 20 software. First, frequencies and medians were used for the basic descriptors, while the associations between the detailed variables was analyzed using independent *t*-test samples and a one-way analysis of variance (ANOVA) depending on the number of categories of each variable. In addition, a one-way multivariate analysis of variance (MANOVA) was performed to examine whether the results for TM, MD, and MET-minutes/week, taken together as three dependent variables, were determined by the country of origin, Romania or Spain. Levene’s test was used to check homoskedasticity. A correlational value above 0.5 was considered to be strong, values between 0.3–0.49 were considered moderate, and any value less than 0.29 was considered to be poor [33].

## 3. Results

### 3.1. Differences between Dependent Variables (H1)

The results from TM (time management, with a detailed presentation on the three dimensions of the variable and on the indicators related to each dimension), PA, and MD based on the country of origin of the respondents have been summarized in Table 1.

For the TM domain, our analysis showed that there were no differences between the respondents from the two countries in their overall AM, AP, SA, or TM scores. Significant differences occurred only in some of the variable size indicators. At the “activity planning” (PA) dimension, only one indicator out of the four, i.e., the distribution of time on the planned activities, registered a significant difference (*p* = 0.015; 6.98 ± 1.62 vs. 6.45 ± 2.02), with the Romanians obtaining a higher score. In terms of activity monitoring (MA), all three indicators showed significant differences, but when using the work agenda (3.69 ± 1.04 vs. 3.40 ± 1.05) and redistributing the time (6.77 ± 1.96 vs. 6.02 ± 1.74), Romanians scored higher, while for the observance of the time allocated to activities (11.59 ± 2.55 vs. 12.94 ± 2.11) the Spanish recorded an increased score. This aspect that balanced the final score of the MA, with the difference on this dimension being insignificant (*p* = 0.548). A similar situation occurred with the size of the efficiency of activities (SA), where the indicator, “call for support” (6.38 ± 1.62 vs. 5.78 ± 1.84), the Romanians obtained a higher score, while Spanish had a higher score for “making use of free slots of time” (6.23 ± 2.42 vs. 6.88 ± 2.22), an aspect that counterbalances the total final SA score, determining the non-existence of a significant difference (*p* = 0.399) between the Romanian and Spanish respondents.

As can be seen in Table 1, regarding the variable PA-MET_minutes, the difference was significant (*p* ≤ 0.001, *p* = 0.001) between Romanians and Spanish (3390.51 ± 3111.57 vs. 4683.32 ± 2823.85) with Spanish students obtaining a higher score. The variable MD also showed a significant difference (*p* ≤ 0.001, *p* = 0.001) between Romanian and Spanish (4.23 ± 2.95 vs. 6.38 ± 2.41) students, with respondents from Spain having a higher score.

Table 2 presents, using an independent-samples *t*-test, the results obtained for TM (the time management presentation on the three dimensions of the variable and on the indicators related to each dimension), PA, and the MD, depending on the faculty of origin of the respondents.

The TM variable did not show a significant difference depending on the faculty from which the respondents came. Significant differences between sports students and the psychology and education sciences students appeared only in one of the indicators of the “activity efficiency” (SA) dimension, even if the SA size did not register a significant difference (*p* = 0.413) between the respondents from the two faculties. There was a significant difference in the “call for support” indicator at *p* < 0.05 (*p* = 0.013), with students in psychology and educational sciences obtaining an increased score (5.97 ± 1.72 vs. 6.47 ± 1.68).

The variable PA, as expected, registered a strong significant difference for students from sports to MET_minutes (*p* = 0.001). They registered increased values compared to those from psychology and education sciences, both at MET_vigorous (*p* = 0.001; 2537.31 ± 2209.00 vs. 882.24 ± 1395.96), and MET_moderate (*p* = 0.038; 810.18 ± 921.57 vs. 575.52 ± 998.11) levels. Regarding the MD variable, the difference was significant for sports students (*p* ≤ 0.001; *p* = 0.001), who recorded a higher score (5.47 ± 2.85 vs. 4.21 ± 2.96).

Table 3 presents, using an independent-samples *t*-test, the results obtained for TM (the time management presentation on the three dimensions of the variable and on the indicators related to each dimension), PA, and MD, according to gender.

The variable, TM, does not register a significant difference according to gender (*p* = 0.853). Significant differences between women and men appeared in only two of the indicators of the “activity efficiency” (SA) dimension, but the size of the SA did not show a significant difference (*p* = 0.782) between female and male respondents. There was a statistically significant difference in the “call for support” indicator from *p* ≤ 0.001 (*p* = 0.003), with female students scoring higher (6.40 ± 1.67 vs. 5.80 ± 1.73). Moreover, for the indicator “making use of free slots of time” there was a significant difference at *p* < 0.05 (*p* = 0.020), this time with the male students obtaining an increased score (6.20 ± 2.43 vs. 6.87 ± 2.21).

According to the data in Table 4, PA showed a strong significant difference for male students in the MET_minutes (*p* < 0.001; *p* = 0.002), which recorded higher values compared to female students (3394.45 ± 2999.99 vs. 4556.75 ± 3079.51). In MET_vigorous activity (*p* = 0.001; 1333.40 ± 1852.58 vs. 2717.41 ± 2152.61) males registered increased values, and in MET_easy activity (*p* = 0.038; 810.18 ± 921.57 vs.575.52 ± 998.11) females obtained a higher score. Regarding the MD variable, the difference was not significant between the values obtained by the genders (4.75 ± 3.02 vs. 5.27 ± 2.85).

Table 4 indicates the dimensions (AP, AM, and SA) of the TM variable, the MD, and the levels of PA, low, moderate, or high. According to the data presented, there were no significant differences depending on the level of PA.

Table 5 indicates the dimensions (AP, AM, and SA) of the TM variable and the PA variable according to the levels of MD, low, moderate, or high. It was observed that the significant differences were found in the dimensions of the planned activity (*p* = 0.024) and variable PA (*p* = 0.006). Thus, it was observed that those who had a high MD planned their time management activities better (38.42 ± 4.97 vs. 36.01 ± 6.24 and 36.15 ± 6.11) and did more intense PA (4842.65 ± 2968.55 vs. 3861.04 ± 3531.05 and 3347.46 ± 2698.15).

### 3.2. Correlations between Variables (H2)

Table 6 shows the correlations between the variables studied. A small positive relationship was observed between TM and MD (*p* < 0.01; r = 0.16) and TM and MET_vigorous activity (*p* < 0.05; r = 0.12). Moreover, a positive association was noted between the MD and MET_vigorous activity (*p* < 0.01; r = 0.15). The dimensions of time management were all interconnected, indicating a high positive correlation (the r value ranged between 0.53 and 0.6). The correlation between MET_vigorous, MET_moderate, and MET_easy activities ranged from small to large (r = 0.1–0.85).

Following the results of the relationships between the variables (Table 7) on the component subgroups of the investigated group, some significant aspects were found depending on the country of origin, the study program, and gender. Thus, it is noted that, regardless of the country of origin or gender, in the students from the psychology and education sciences faculty, there were no correlations between any of the variables.

Spanish female students enrolled in the sports program showed moderate-to-strong values for TM and the MD (r = 0.60; *p* < 0.01), TM and MET_minutes (r = 0.47; *p* < 0.01), and MD and MET_minutes (r = 0.52; *p* < 0.01). With regards to Spanish male students in the sports faculty, significant correlations between TM and the MD (r = 0.30; *p* < 0.05) and the MD and MET_minutes (r = 0.30; *p* < 0.05) were observed.

In the Romanian respondents from the physical education and sports study program, significant correlations appeared as follows: for females, a significant negative correlation, at *p* < 0.05, was observed between MD and MET_minutes (r = −0.36); for males, a strongly significant positive correlation at *p* < 0.01 between TM and MET_minutes (r = 0.41) was observed.

### 3.3. The Determination of Results and the Country Predictor (H3)

A one-way multivariate analysis of variance (MANOVA) was performed to examine whether the results for TM, the MD, and MET_minutes, were determined by the country of origin, Romania or Spain (Table 8). The results indicated that the country (Wilk’s Lambda = 0.944, F(3, 286) = 5.69, *p* = 0.001, *p* < 0.05, partial eta squared = 0.056) had a significant relationship with the combination of dependent variables.

## 4. Discussion

Our study aimed to determine the correlation between TM, the MD, and PA variables in Spanish and Romanian university students. In summary, the findings from this study suggest that there were no overall differences in the TM scores of participants based on the country, gender, and faculty. However, there were differences in the PA score when these parameters were considered. Significant differences were observed when the MD score was based on the country and faculty, but no differences were observed when a gender comparison was performed. A significant relationship could not be determined between TM, the MD, and the various levels of PA. Furthermore, our participants with a higher MD adherence were able to plan activities better and perform more intense PA compared to the other participants. Moreover, a positive relationship was observed between TM and the MD (*p* < 0.01; r = 0.16) and TM and MET_vigorous activity (*p* < 0.05; r = 0.12). Similarly, a small positive association was noted between the MD and MET_vigorous activity (*p* < 0.01; r = 0.15). Finally, the MD and MET_minutes is influenced by the country of the participants. In summary, the results partly support our hypothesis that there is a correlation between TM, PA, and the MD variables, and this can be influenced by the country of the university students. The findings of this study are of relevance to university students involved in the Faculty of Psychology and Educational Sciences, or the Faculty of Physical Education and Sports in Spain and Romania.

Time management is a skill that can be developed at any age, provided the individual is willing to improve the results of their actions [34]. An effective TM strategy is associated with improved academic performance and lower stress levels [35]. Even though the country, faculty, and gender did not influence the overall TM, there were a few significant differences in the individual domains of TM questionnaire. For instance, the time distribution during activity-planning was reported to be better in Romanian students compared to Spanish students. However, it was noted that Spanish students were better at activity-monitoring tasks, such as scheduling, observing the time allocated to each activity, and the redistribution of time. This could be due to various reasons. Firstly, the type of activities for which TM might be of importance were not included in the list of planned activities. Secondly, there is a possibility that these activities were included in the list, but the participating students did not monitor the time allocated to them. Therefore, university students should focus on improving the overall TM qualities in order to improve academic performance, PA, and reduce the perceived stress levels.

Our results indicate that the adherence to the MD diet in Spanish students was higher in comparison to Romanian students. The students were provided meals inside the university campus. This is in line with a previous study in which Spanish students have been reported to have a higher adherence to the MD food compared to their counterparts [24]. One reason for this is due to the geographical location of Romania, which is a non-coastal country and is further north than Spain [25]. This could have limited the cultivation of the food of the Mediterranean basin in Romania [36]. Moreover, the climatic conditions in Spain favor the consumption of fish, milk, and cereals, which could have helped Spanish students to adhere to the MD. Furthermore, we observed that sports students had a higher adherence to the MD compared to psychology students. This could be due to the application of the academic knowledge in nutrition regarding the benefits of the MD to their individual lives. We also observed that better activity planning was associated with an increased adherence to MD. A potential reason could be that the consumption of MD was a priority for the students in both countries and it would have been given priority over other activities.

We found that PA activities varied between the countries, genders, and the faculty. Furthermore, we also observed a strong correlation between the MD and PA, with the former improving the performance of the latter. The PA activities were observed to be higher in Spanish students compared to Romanian students. Young people practicing physical activity tend to consume a nutritious diet to obtain greater results in terms of performance, body image, or wellbeing [37,38]. This could be one potential reason for the adherence to the MD by Spanish university students. Furthermore, male participants were observed to be more physically active than females. This has been observed in previous studies as well [39,40,41]. However, further research needs to be conducted in order to determine the causal effect of such differences in PA depending on the gender of the university students.

There was no significant difference in the TM and the MD based on the various levels of PA. Previous studies have reported that a greater adherence to MD has been associated with more PA [22]. However, such results were not observed in our study. The disparity in findings could be due to the variation in the study design and the participants included in our study. However, our results are in line with the study by Badicu et al. [23] in which no significant relationship was observed between the MD and PA in university students. Therefore, it can be speculated that the participants in our study were engaged in some sort of physical activity. Furthermore, there was no significant difference in the overall TM of the students. This could be one of the reasons for the lack of differences in the PA activities of university students in Spain and Romania. Hence, it can be said that the constructive use of time and prior planning may lead to a prudent use of time and a rise in physical activity.

Our bivariate correlation analysis revealed that overall TM was correlated with the MD and MET_vigorous. A study by Valenzuela et al. [42] reported that PA might help in improving the TM skills of university students, since regular PA requires efficient activity planning. Furthermore, university students have been observed to spend long durations in a sedentary environment in order to cope with the study demands [43]. As a result, PA might be a useful tool for improving concentration [44], intellectual development [45,46], and cardiovascular health improvement. Therefore, engaging in regular PA can be useful in developing the TM skills in university students, as well as improving overall health. However, it is suggested that further research needs to be conducted in students from other departments in order to get a better understanding on this topic.

A higher adherence to the MD was associated with more vigorous physical activity. Although not particularly relevant to the university students, the MD has been associated with improved cardiovascular fitness [47], muscle strength [48], and speed agility in adolescents [49]. This could be due to the high levels of antioxidant characteristics of the MD that could lead to an increased efficiency in oxygen uptake and utilization [50] Furthermore, it may promote the protection of cellular components, such as proteins, from the catabolic effects of oxidative stress that imbalance the relationship between the production of reactive oxygen species and antioxidant defense [51]. Therefore, these could be some of the factors associated with enhanced PA in students adhering to the MD diet. However, further research is required for gaining additional insights into the influence of the MD on PA in university students.

There are a few limitations to our study that need to be highlighted. Firstly, the study was conducted only on students from two departments. A further analysis involving students from other fields of study will help to better understand the correlation between TM, the MD, and PA. Secondly, this study was conducted only on university students. Third, limitations appeared in the sample size and its distribution by country, gender, and faculty. There was double the number of Romanian students in total, and the dominant number of Romanian students were in psychology, compared to the small number of students in psychology in Spain, with a dominant female number. Therefore, the findings cannot be transferred to the general populations in both the countries.

## 5. Conclusions

Our study aimed to determine the correlation between the variables TM, the MD, and PA in Spanish and Romanian university students. There were significant differences observed in the MD and PA parameters, as well as between the Romanian and Spanish respondents, between the respondents from the sports faculty and those from the psychology faculty, and between the female and male respondents. The TM variable did not show a significant difference depending on the country, faculty and gender. There was only a significant relationship between the MD and the other two variables (TM and PA). However, further research needs to be conducted in order to develop a better understanding on this topic.

## Figures and Tables

**Table 1 ijerph-19-02554-t001:** Time management, physical activity, and Mediterranean diet according to country.

					Levene’s Test		*t*-Test	
		Country	M	SD	F	Sig.	*t*	Sig.
Activity planning (AP)	Making a schedule	Romania	7.41	1.63	2.337	0.127	−0.775	0.439
		Spain	7.56	1.48				
	Prioritizing activities	Romania	15.23	3.11	1.157	0.283	1.367	0.173
		Spain	14.72	2.77				
	Time distribution	Romania	6.98	1.62	7.475	0.007	2.459	0.015 *
		Spain	6.45	2.02				
	Clearly-defined goals	Romania	7.26	1.58	0.037	0.848	0.143	0.887
		Spain	7.23	1.57				
	AP total score	Romania	36.89	6.19	1.771	0.184	1.244	0.214
		Spain	35.97	5.54				
Activity monitoring (AM)	Using a schedule	Romania	3.69	1.04	0.095	0.758	2.275	0.024 *
		Spain	3.40	1.05				
	Observing the time allotted to each activity	Romania	11.59	2.55	3.461	0.064	−4.513	0.001 *
		Spain	12.94	2.11				
	Redistributing time	Romania	6.77	1.96	1.666	0.198	3.210	0.001 *
		Spain	6.02	1.74				
	AM total score	Romania	22.06	4.35	5.964	0.015	−0.601	0.548
		Spain	22.36	3.41				
Streamlining activities (SA)	Achieving the activities	Romania	11.46	2.16	0.093	0.760	1.843	0.066
		Spain	10.97	2.12				
	Asking for support	Romania	6.38	1.62	2.732	0.099	2.900	0.004 *
		Spain	5.78	1.84				
	Making use of free slots of time	Romania	6.23	2.42	1.945	0.164	−2.215	0.028 *
		Spain	6.88	2.22				
	SA total score	Romania	24.08	4.45	1.666	0.198	0.845	0.399
		Spain	23.62	4.12				
TM total score	AP + AM + SA	Romania	83.02	13.21	7.214	0.008	0.697	0.486
		Spain	81.95	10.69				
PA	MET_vigorous	Romania	1374.55	1894.71	3.172	0.076	−5.759	0.001 *
		Spain	2775.51	2113.08				
	MET_moderate	Romania	664.34	1031.63	1.365	0.244	−1.191	0.234
		Spain	805.51	792.48				
	MET_easy	Romania	1351.62	1227.43	5.011	0.026	1.729	0.085
		Spain	1102.30	1036.04				
	MET_minutes	Romania	3390.51	3111.57	0.445	0.505	−3.466	0.001 *
		Spain	4683.32	2823.85				
MD	MD total score	Romania	4.23	2.95	6.993	0.009	−6.236	0.001 *
		Spain	6.38	2.41				

Note: *, *p* < 0.05; MET-Metabolic Equivalent of Task; M, Mean; SD, standard deviation; F, F-value; Sig., *p* value; *t*, *t*-value.

**Table 2 ijerph-19-02554-t002:** Time management, physical activity, and Mediterranean diet according to faculty.

					Levene’s Test		*t*-Test	
		Fac.	M	SD	F	Sig.	*t*	Sig.
AP	Making a schedule	Sport	7.51	1.55	0.434	0.511	0.624	0.533
		Psyho	7.39	1.63				
	Prioritizing activities	Sport	15.09	2.70	6.953	0.009	0.157	0.875
		Psyho	15.03	3.39				
	Time distribution	Sport	6.80	1.85	0.133	0.716	−0.071	0.944
		Psyho	6.82	1.68				
	Clearly-defined goals	Sport	7.40	1.51	0.056	0.812	1.843	0.066
		Psyho	7.06	1.65				
	AP total score	Sport	36.80	5.52	4.899	0.028	0.708	0.480
		Psyho	36.30	6.58				
AM	Using a schedule	Sport	3.62	0.97	5.596	0.019	0.482	0.630
		Psyho	3.56	1.15				
	Observing the time allotted to each activity	Sport	12.33	2.33	4.271	0.040	2.358	0.019 *
		Psyho	11.64	2.66				
	Redistributing time	Sport	6.49	1.95	0.438	0.509	−0.399	0.690
		Psyho	6.58	1.89				
	AM total score	Sport	22.43	3.70	4.936	0.027	1.377	0.170
		Psyho	21.78	4.48				
SA	Achieving the activities	Sport	11.19	2.05	1.440	0.231	−1.025	0.306
		Psyho	11.45	2.30				
	Asking for support	Sport	5.97	1.72	0.045	0.832	−2.499	0.013 *
		Psyho	6.47	1.68				
	Making use of free slots of time	Sport	6.59	2.31	0.577	0.448	1.227	0.221
		Psyho	6.25	2.44				
	SA total score	Sport	23.75	4.03	3.759	0.053	−0.820	0.413
		Psyho	24.17	4.73				
TM total score	AP + AM + SA	Sport	82.98	11.05	10.521	0.001	0.503	0.615
		Psyho	82.24	14.141				
PA	MET_vigorous	Sport	2537.31	2209.00	32.214	0.000	7.368	0.001 **
		Psyho	882.24	1395.96				
	MET_moderate	Sport	810.18	921.57	0.010	0.920	2.089	0.038 *
		Psyho	575.52	998.11				
	MET_easy	Sport	1190.80	1157.71	0.334	0.564	−1.346	0.179
		Psyho	1376.15	1186.81				
	MET_minutes	Sport	4538.28	3211.74	4.431	0.036	4.889	0.001 **
		Psyho	2833.91	2582.53				
MD	MD total score	Sport	5.47	2.85	0.340	0.560	3.704	0.001 **
		Psyho	4.21	2.96				

Note: *, *p* < 0.05; **, *p* < 0.01; AP, activity planning; AM, activity monitoring, SA, streamlining activities; PA, physical activity; MD, Mediterranean diet; M, mean; SD, standard deviation; Sig., *p* value; F, F-value; *t*, *t*-value.

**Table 3 ijerph-19-02554-t003:** Time management, physical activity, and Mediterranean diet, according to sex.

					Levene’s Test		*t*-Test	
		Sex	M	SD	F	Sig.	*t*	Sig.
AP	Making a schedule	Female	7.34	1.58	0.163	0.687	−1.785	0.075
		Male	7.68	1.57				
	Prioritizing activities	Female	15.31	3.12	2.548	0.112	1.889	0.060
		Male	14.63	2.77				
	Time distribution	Female	6.78	1.74	0.052	0.821	−0.393	0.695
		Male	6.86	1.85				
	Clearly-defined goals	Female	7.22	1.61	0.070	0.791	−0.430	0.668
		Male	7.31	1.52				
	AP total score	Female	36.65	6.08	0.852	0.357	0.244	0.807
		Male	36.47	5.85				
AM	Using a schedule	Female	3.61	1.11	6.189	0.013	0.368	0.713
		Male	3.56	0.94				
	Observing the time allotted to each activity	Female	11.91	2.54	0.295	0.587	−1.112	0.267
		Male	12.25	2.40				
	Redistributing time	Female	6.41	1.93	0.129	0.719	−1.345	0.180
		Male	6.72	1.90				
	AM total score	Female	21.94	4.27	2.621	0.107	−1.227	0.221
		Male	22.54	3.63				
SA	Achieving the activities	Female	11.37	2.25	2.129	0.146	0.787	0.432
		Male	11.17	2.00				
	Asking for support	Female	6.40	1.67	0.135	0.714	2.967	0.003 **
		Male	5.80	1.73				
	Making use of free slots of time	Female	6.20	2.43	1.720	0.191	−2.349	0.020 *
		Male	6.87	2.21				
	SA total score	Female	23.98	4.49	2.672	0.103	0.277	0.782
		Male	23.83	4.08				
TM total score	AP + AM + SA	Female	82.56	13.12	6.471	0.011	−0.185	0.853
		Male	82.84	11.17				
PA	MET_vigorous	Female	1333.40	1852.58	5.358	0.021	−5.827	0.001 *
		Male	2717.41	2152.61				
	MET_moderate	Female	653.51	1025.92	1.529	0.217	−1.363	0.174
		Male	811.30	828.02				
	MET_easy	Female	1407.54	1254.78	11.314	0.001	2.711	0.007 **
		Male	1028.04	970.03				
	MET_minutes	Female	3394.45	2999.99	0.076	0.784	−3.178	0.002 **
		Male	4556.75	3079.51				
MD	MD total score	Female	4.75	3.02	0.277	0.599	−1.451	0.148
		Male	5.27	2.85				

Note: *, *p* < 0.05; **, *p* < 0.01; AP, activity planning; AM, activity monitoring, SA, streamlining activities; PA, physical activity; MD, Mediterranean diet; M, mean; SD, standard deviation; F, F-value; *t*, *t*-value; Sig., *p* value; Female *n* = 188, Male *n* = 108.

**Table 4 ijerph-19-02554-t004:** The relationship between time management, Mediterranean diet, and adherence to physical activity.

PA		*n*	M	SD	F	*p*
AP	Low	48	34.88	6.47	2.512	0.083
	Moderate	118	37.14	6.09		
	High	130	36.72	5.63		
AM	Low	48	21.33	4.52	1.274	0.281
	Moderate	118	22.19	4.10		
	High	130	22.42	3.82		
SA	Low	48	23.06	4.33	2.984	0.052
	Moderate	118	24.64	4.37		
	High	130	23.59	4.25		
TM total score	Low	48	79.27	13.76	2.470	0.086
	Moderate	118	83.97	12.56		
	High	130	82.73	11.63		
MD	Low	48	4.79	2.96	2.409	0.092
	Moderate	118	4.54	2.87		
	High	130	5.35	3.01		

Note: AP, activity planning; AM, activity monitoring, SA, streamlining activities; PA, physical activity; MD, Mediterranean diet; M, mean; SD, standard deviation; F, F-value; *p*, *p* value.

**Table 5 ijerph-19-02554-t005:** Relationship between time management, physical activity, and adherence to Mediterranean diet.

MD		*n*	M	SD	F	*p*
AP	Low	91	36.01	6.24	3.766	0.024 *
	Moderate	143	36.15	6.11		
	High	62	38.42	4.97		
AM	Low	91	21.65	4.45	1.087	0.338
	Moderate	143	22.31	3.79		
	High	62	22.53	4.04		
SA	Low	91	23.77	4.48	0.610	0.544
	Moderate	143	23.79	4.51		
	High	62	24.47	3.70		
TM total score	Low	91	81.43	13.57	2.063	0.129
	Moderate	143	82.26	12.49		
	High	62	85.42	10.10		
PA	Low	91	3861.04	3531.05	5.272	0.006 **
	Moderate	143	3347.46	2698.15		
	High	62	4842.65	2968.55		

Note: *, *p* < 0.05; **, *p* < 0.01; AP, activity planning; AM, activity monitoring, SA, streamlining activities; PA, physical activity; MD, Mediterranean diet; M, mean; SD, standard deviation; F, F-value; *p*, *p* value.

**Table 6 ijerph-19-02554-t006:** Bivariate correlations between the variables and dimensions of TM.

	AM	SA	TM	MD	MET_v	MET_m	MET_e	MET_t
AP	0.685 **	0.602 **	0.916 **	0.177 **	0.094	0.021	0.084	0.102
AM		0.530 **	0.842 **	0.132 *	0.148 *	−0.019	−0.021	0.086
SA			0.813 **	0.088	0.060	−0.057	−0.029	0.012
TM (AP + AM + SA)				0.159 **	0.115 *	−0.016	0.023	0.081
MD					0.151 **	0.070	−0.034	0.111
MET_vigorous						0.407 **	0.135 *	0.853 **
MET_moderate							0.261 **	0.686 **
MET_easy								0.553 **

AP, activity planning; AM, activity monitoring, SA, streamlining activities, TM, time management; MET, physical activity minutes; MD, Mediterranean diet; ** correlation is significant at the 0.01 level (two-tailed); * correlation is significant at the 0.05 level (two-tailed).

**Table 7 ijerph-19-02554-t007:** Bivariate correlations between the variables, by subgroups.

C	Faculty	Sex		MD Total Score	MET_Minutes
RO *n* = 198	S *n* = 85	F *n* = 45	TM total score	0.284	−0.065
			MD total score		−0.355 *
		M *n* = 40	TM total score	0.179	0.406 **
			MD total score		0.157
	P *n* = 113	F *n* = 106	TM total score	0.103	0.030
			MD total score		−0.084
		M *n* = 7	TM total score	−0.109	0.510
			MD total score		−0.334
SP *n* = 98	S *n* = 86	F *n* = 29	TM total score	0.597 **	0.471 **
			MD total score		0.516 **
		M *n* = 57	TM total score	0.301 *	−0.152
			MD total score		0.296 *
	P *n* = 12	F *n* = 8	TM total score	−0.201	0.132
			MD total score		0.627
		M *n* = 4	TM total score	−0.279	0.592
			MD total score		0.597

** Correlation is significant at the 0.01 level (two-tailed); * correlation is significant at the 0.05 level (two-tailed); C = country; RO = Romania; SP = Spain; S = Sport; P = Psychology; F = Female; M = Male; TM = Time Management; MD = Mediterranean diet; MET-minute = activity volume in minutes.

**Table 8 ijerph-19-02554-t008:** Multivariate test ^a^ of variance between the three dependent variables and the country.

Effect		Value	F	Hypothesis DF	Error DF	Sig.	Partial Eta Squared	Noncent. Parameter	Observed Power ^c^
Intercept	Pillai’s Trace	0.941	1524.733 ^b^	3.000	286.000	0.000	0.941	4574.199	1.000
	Wilks’ Lambda	0.059	1524.733 ^b^	3.000	286.000	0.000	0.941	4574.199	1.000
	Hotelling’s Trace	15.994	1524.733 ^b^	3.000	286.000	0.000	0.941	4574.199	1.000
	Roy’s Largest Root	15.994	1524.733 ^b^	3.000	286.000	0.000	0.941	4574.199	1.000
Country	Pillai’s Trace	0.056	5.699 ^b^	3.000	286.000	0.001	0.056	17.098	0.946
	Wilks’ Lambda	0.944	5.699 ^b^	3.000	286.000	0.001	0.056	17.098	0.946
	Hotelling’s Trace	0.060	5.699 ^b^	3.000	286.000	0.001	0.056	17.098	0.946
	Roy’s Largest Root	0.060	5.699 ^b^	3.000	286.000	0.001	0.056	17.098	0.946

^a^. Design: Intercept + country; ^b^. Exact statistic; ^c^. Computed using alpha = 0.05.

## Data Availability

The data used to support the findings of current study are available from the corresponding author upon request.

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
