# Peer review of "Exploring the Correlation between Time Management, the Mediterranean Diet, and Physical Activity: A Comparative Study between Spanish and Romanian University Students"

_ijerph, 2022, doi:10.3390/ijerph19052554_

Round 1

Reviewer 1 Report

Comments to the Author:

I thank the editors for the opportunity to review this study and congratulate the authors for the effort put into their study. This manuscript by Indreica discussed "Exploring the Correlation between Time Management, Mediterranean Diet and Physical activity: A Comparative Study Between Spanish and Romanian University Students". The authors intended the purpose of this study to conduct a comparative study between sports and psychology students - education sciences, from Romania and Spain. This study lacks novelty, it is quite difficult to understand and identify what the authors want to demonstrate with it.

  1. I think that the numbering in the abstract is in a mistake

  1. First, the authors must justify or add the current existing problem that makes necessary to carry out this study. Different questions arise for me:

  1. Why should it be Spain and not Italy?
  2. What is the significance of this comparison?
  3. What knowledge will this project bring to science? And what relevance does it have?
  4. Why was it used for students rather than the general adult population?

  1. On the other hand, the introduction needs to be better worked out and adds too little information about the relevance of the study.
  2. The main problem with the study is that it does not have a sufficient sample size for the data to be valid and reliable, indeed, there is a big difference between the Spanish and Romanian samples.
  3. Were the test used validated in the languages in which they were used, i.e. Spanish and Romanian?
  4. This study cannot be extrapolated to the university community, because the areas chosen are only two (Physical education and sports and 125 student from psychology-educational sciences). Therefore, it is not clear why the questionnaire was not extended to cover a larger number of universities so that the results could be extrapolated.

Author Response

Dear Reviewer, 

I'm attaching the changes.

Thank you!

Reviewer 2 Report

In my opinion, the idea and objective of the paper is good. But some issues need to be addressed:

  • the introduction need some more information and references to frame the work that is later described
  • this is a study on volunteers, so the sample that was recruited can be very biased compared with the general population. Moreover, part of the students were engaged in a physical education, probably biasing even more the sample, since it is more probably that these students have higher PA levels and also probably better diet
  • Regarding the evaluation of MD, the authors used the KIDMED questionnaire, which is originally designed to assess the Mediterranean diet in children and adolescents. A different tool, more oriented to adults would be more appropiate
  • There is no description about the ethics approval for the study in Romania, only for the Spanish study
  • Results: I would encourage the authors to use some multivariate analyses, in order to take into account different variables at the same time. Authors show only bivariate analyses.

Author Response

(The authors gave the same response as above.)

Reviewer 3 Report

Title

The title of the paper is very accurate and clear presentation of the content of the manuscript.

Abstract

Overall, the abstract is well written, I have only some small concerns.

I am not sure if the sample size of 296 is great enough to say that this is a fact-finding investigation. Also, the conclusions of abstract should be more specific.

Introduction

Overall, the introduction is clear and well written, but on the other hand it is way too short. For example, please check line 33 – is there anything missing?

The Authors should greatly broaden the introduction section. For example, what are the key correlates of (1) Mediterranean diet, and (2) physical activity – Authors completely miss such kind of information. For example, recent studies clearly show that the intrinsic motivation is a key correlate of daily moderate-to-vigorous physical activity (Kalajas-Tilga et al., 2020).

Kalajas-Tilga, H., Koka, A., Hein, V., Tilga, H., & Raudsepp, L. (2020). Motivational processes in physical education and objectively measured physical activity among adolescents. Journal of Sport and Health Science, 9(5), 462–471. https://doi.org/10.1016/j.jshs.2019.06.001

And to improve the introduction about Mediterranean diet you might want to check the study by Galan-Lopez et al. (2020).

Galan-Lopez, P., Sanchez-Oliver, A. J., Pihu, M., Gísladóttír, T., Domínguez, R., & Ries, F. (2020). Association between Adherence to the Mediterranean Diet and Physical Fitness with Body Composition Parameters in 1717 European Adolescents: The AdolesHealth Study. Nutrients, 12(1), 77. https://doi.org/10.3390/nu12010077

The introduction ends with a clear statement of the aim and hypotheses of the study. However, I suggest Authors to make a separate paragraph entitled “The present study”.

Materials and Methods – clear and accurate

Results – mostly clear

Please check the tables – what does the “sig” stand for”? Is it the significance? If yes, the sig value cannot be equal to zero, it is always different from zero. Also, I cannot understand how sig value can have another *p-value? Isn’t it overlapping?

In table 4, specifically in the note the p-value is defined but the p value is also in the table – thus, it is overlapping. Also, Authors do not use *-s in that table, so it should not be in the note.

Table 5 – the p-value has its own p-value – I believe this must be a typo? Please revise.

Discussion

There are so many results presented in this study, but the discussion is only 1 page long. The discussion needs extensive revisions.

Author Response

(The authors gave the same response as above.)

Reviewer 4 Report

Dear authors,

The topic of the manuscript is interesting but it needs some improvements for better comprehension.

Abstract

the objective is not clear, as well as at the end of the introduction section. Author start the discussion section with a clear objective of the study. therefore, change in the abstract and in the introduction.

Improve the conclusion in the abstract.

Introduction

It is confusing - a mix of introduction and methods on the first paragraph.

Lines 54-57 - No references to support the information. It does not belong in the introduction section

Better define TM in the introduction. Do the articles relate TM and PA?

Materials

Why do you have a double number of Romanian students?

Were all the students from the faculty invited? How?

All the descriptions of the students belong to the results section.

How much time was it necessary for the student to answer the whole research instruments?

Kidmed is for youth, what about the older students in the sample?

Line 121 - change additional to additionally. and insert "it" before was

Were the original instruments already in Romanian or Spanish? Were they translated and culturally adapted?

Results

sentence 171 - take off the text

line 251 - change too for to

Discussion

From lines 268 to 281, authors present results of the study. there is no discussion.

Line 293 - it brings a discussion that the students should focus on improving overall TM, but these results were not presented for a discussion.

307 - 308 - in both countries

More discussion is needed to explore countries' differences of MD and PA. Does the university have sports program? Do students have meals inside campus?

Relate TM, PA and MD during covid-19

Conclusion 

Improve based on the results

Author Response

(The authors gave the same response as above.)

Round 2

Reviewer 1 Report

Thanks to the authors for their efforts, however, I am sorry to inform the authors that the modifications made by the authors are not sufficient to solve the main problems of the manuscript. I send you my best regards and hope that in future studies you will take into account my recommendations above. 

Author Response

Dear Reviewer, 

Thanks to the authors for their efforts, however, I am sorry to inform the authors that the modifications made by the authors are not sufficient to solve the main problems of the manuscript. I send you my best regards and hope that in future studies you will take into account my recommendations above.

Response: Thanks to the reviewer for the evaluation. From the ones presented by the reviewer at the previous revision, we mention that, the authors made all the changes point by point, as requested. For the next stage, the reviewer presented a sentence with the general comments without referring to what was not done, or was not answered properly, by the authors.

Thank you!

Reviewer 3 Report

Authors have done well job on revising the manuscript.

Author Response

Dear Reviewer, 

Authors have done well job on revising the manuscript.

Response: Thank you very much for assessments.

Reviewer 4 Report

Dear authors, much improvement has been done in the manuscript, but some minor errors need to be corrected.

line 17, change to: the aim of our study is to determine the correlation between the variables TM, MD and PA in Spanish and Romanian university students. 

line 37 -How, following Following  (rewrite)

Include in the methodology the average time for a student to respond the search and also include a sentence explaining that the original instruments were used in Romanian and Spanish as other studies already used them in these languages.

line 362 - take out the from both the countries, line 364 as well

line 370 - MET v. - correct

line 373 - take out ab

Relate TM, PA and MD during covid-19 as requested before. This was not included in the discussion section.

Author Response

Dear Reviewer, 

Thank you!
